# Effects of 6,8-Diprenylgenistein on VEGF-A-Induced Lymphangiogenesis and Lymph Node Metastasis in an Oral Cancer Sentinel Lymph Node Animal Model

**DOI:** 10.3390/ijms22020770

**Published:** 2021-01-14

**Authors:** Mun Gyeong Bae, Jeon Hwang-Bo, Dae Young Lee, Youn-Hyung Lee, In Sik Chung

**Affiliations:** 1Department of Genetic Engineering and Graduate School of Biotechnology, Kyung Hee University, Yongin 446-701, Korea; max_angel@naver.com (M.G.B.); hbj3286@khu.ac.kr (J.H.-B.); 2Department of Herbal Crop Research, National Institute of Horticulture and Herbal Science, RDA, Eumseong 27709, Korea; dylee0809@gmail.com; 3Department of Horticultural Biotechnology, Kyung Hee University, Yongin 17104, Korea; younlee@khu.ac.kr

**Keywords:** oral cancer, 6,8-diprenylgenistein, VEGF-A-induced lymphangiogenesis, sentinel lymph node metastasis

## Abstract

Background: The major determining factor of prognosis of oral squamous cell carcinoma is cervical lymph node metastasis. 6,8-Diprenylgenistein (6,8-DG), an isoflavonoid isolated from *Cudrania tricuspidata* has been reported to have anti-microbial and anti-obesity activities. However, its effects on lymphangiogenesis and lymph node metastasis in oral cancer have not yet been reported. Methods: To investigate the in vitro inhibitory effects of 6,8-DG on VEGF-A-induced lymphangiogenesis, we performed the proliferation, tube formation, and migration assay using human lymphatic microvascular endothelial cells (HLMECs). RT-PCR, Western blot, immunoprecipitation, ELISA and co-immunoprecipitation assays were used to investigate the expression levels of proteins, and mechanism of 6,8-DG. The in vivo inhibitory effects of 6,8-DG were investigated using an oral cancer sentinel lymph node (OCSLN) animal model. Results: 6,8-DG inhibited the proliferation, migration and tube formation of rhVEGF-A treated HLMECs. In addition, the in vivo lymphatic vessel formation stimulated by rhVEGF-A was significantly reduced by 6,8-DG. 6,8-DG inhibited the expression of VEGF-A rather than other lymphangiogenic factors in CoCl_2_-treated SCCVII cells. 6,8-DG inhibited the expression and activation of VEGFR-2 stimulated by rhVEGF-A in HLMECs. Also, 6,8-DG inhibited the activation of the lymphangiogenesis-related downstream signaling factors such as FAK, PI3K, AKT, p38, and ERK in rhVEGF-A-treated HLMECs. Additionally, 6,8-DG inhibited the expression of the hypoxia-inducible factor (HIF-1α), which is involved in the expression of VEGF-A in CoCl_2_-treated SCCVII cells, and 6,8-DG inhibited VEGF-A signaling via interruption of the binding of VEGF-A and VEGFR-2 in HLMECs. In the VEGF-A-induced OCSLN animal model, we confirmed that 6,8-DG suppressed tumor-induced lymphangiogenesis and SLN metastasis. Conclusion: These data suggest that 6,8-DG inhibits VEGF-A-induced lymphangiogenesis and lymph node metastasis in vitro and in vivo. Furthermore, the inhibitory effects of 6,8-DG are probably mediated by inhibition of VEGF-A expression in cancer cells and suppression of the VEGF-A/VEGFR-2 signaling pathway in HLMEC. Thus, 6,8-DG could be novel and valuable therapeutic agents for metastasis prevention and treatment of oral cancer.

## 1. Introduction

Oral cancer, a type of head and neck cancer, is a malignant tumor of the tongue, gums and oral cavity. Most of the oral cancer is known as squamous cell carcinoma. The main treatment for oral squamous cell carcinoma (OSCC) is surgery, radiation, and chemotherapy, followed by radiation therapy and chemotherapy after surgery by single treatment or combination therapy. However, despite these treatments, only 30% of OSCC patients survive for more than five years [1]. So, new treatment methods for curing OSCC are required.

The main factor that affects the prognosis of patients with OSCC is regional lymph node metastasis. It primarily occurs via the sentinel lymph node (SLN), the first lymph node draining from the primary tumor. Sentinel lymph nodes proximate to the primary tumors are commonly enlarged because of reactive lymphadenopathy, tumor metastasis, or both, suggesting that LN alteration results from interactions between tumors and the lymphatic system [2,3]. The SLN metastasis is an important predictor of prognosis and treatment for oral cancer patients. Also, according to recent reports, the lymphatic system is a more important route than the vascular system in metastasis of oral cancer [4,5].

Lymphangiogenesis, the essential process of lymph node metastasis, is the process of new lymphatic vessel formation from an existing lymphatic vessel, which occurs during tissue repair, inflammation, tumor growth, and tumor metastasis [6]. Lymphangiogenesis plays an important role in promoting the migration of tumor cells from the primary tumor to the SLN. Recently, several studies have reported that malignant tumors promote lymphangiogenesis in SLNs before metastasis [2,7]. Lymphatic vessel density of intra- or peri-tumor is an important parameter for assessing the malignant potential of tumors and patient survival [8]. Tumor-induced SLN lymphangiogenesis occurs before metastasis in OSCC, and tumor-derived VEGF-A and VEGF-D play significant roles in that process [3]. Several studies have reported that tumor lymphangiogenesis correlates with lymph node metastasis and is a prognostic indicator for OSCC, melanoma, and non-small cell lung cancer [8,9,10]. The contribution of the lymphatic system to tumor-related lymph node metastasis is being increasingly appreciated through studies of human cancer tissues, such as carcinoma of the breast, oral cavity, colon, and prostate as well as melanoma [3].

Tumor-induced lymphangiogenesis is mediated by lymphangiogenic growth factors, which are secreted by the tumors [2]. Several studies have shown that increased lymphangiogenesis and lymphangiogenic growth factor promote lymph node metastasis and poor prognosis [6,11]. There are the VEGF family, PDGFs, IGF, FGF, HGF, angiopoietin as lymphatic growth factors and VEGF-C and -D are known as major lymphangiogenic factors involved in tumor-induced lymphangiogenesis in many studies [12,13,14]. In recent studies, VEGF-A as well as VEGF-C and VEGF-D have been reported to act as lymphangiogenic factors on tumor-induced lymphangiogenesis and lymph node metastasis [15,16].

Generally, VEGF-A has been known to be an angiogenic key factor, and promotes angiogenesis via binding to its receptors VEGFR-1 and VEGFR-2 in vascular endothelial cells [15,17]. Recently, it has been reported that VEGF-A may be involved in mediating tumor-induced lymphangiogenesis and lymph node metastasis [15]. VEGFR-2 has higher affinity towards VEGF-A, -E, and lower affinity for VEGF-C, -D. The binding of VEGF-A to the extracellular domain of VEGFR-2 activates the autophosphorylation of the tyrosine residue of VEGFR-2, activating the signaling pathway involved in endothelial cell survival [18].

Isoflavonoids are a class of flavonoid phenolic compounds distributed only in plants and mainly synthesized in soybeans and other leguminous plants. The most important isoflavones are genistein and daidzein [19]. In particular, genistein, a dietary soybean component, has been shown to inhibit prostate tumor growth and angiogenesis [20]. In addition, epigallocatechin-3-gallate (EGCG) and epigallocatechin (EGC), which are polyphenol flavonoids in green tea, have been shown to inhibit angiogenesis both in vitro and in vivo [21,22].

6,8-Diprenylgenistein (6,8-DG) is a major isoflavonoid isolated from *Cudrania tricuspidata* fruit. It has been reported that *C. tricuspidata* extracts have anti-tumor, anti-inflammatory, and anti-oxidant activities [23,24,25]. 6,8-DG has anti-obesity activity in high fat diet-induced obese mice [26], and 6,8-DG has anti-microbial activity against Streptococcus mutants [27]. However, the effects of 6,8-DG on lymphangiogenesis and lymph node metastasis in oral cancer have not yet been reported.

In this study, we investigated the inhibitory effect of 6,8-DG on VEGF-A-induced lymphangiogenesis and lymph node metastasis using in vitro assays and an OCSLN animal model. We further examined the mechanism involved in the inhibitory action of 6,8-DG.

## 2. Results

### 2.1. Effect of 6,8-Diprenylgenestein on VEGF-A-Induced Proliferation, Tube Formation, and Migration of HLMECs

Proliferation, migration, and tube formation of lymphatic endothelial cells are the major steps in lymphangiogenesis. We have confirmed in our previous study that VEGF-A is one of the major lymphangiogenic factors in hypoxic oral cancer cells [7]. Thus, proliferation, migration, and tube formation assays were performed using rhVEGF-A treated HLMECs to confirm the effect of 6,8-DG on oral cancer-related lymphangiogenesis in vitro. The cell density of HLMECs treated with 20 ng/mL rhVEGF-A for 48 h was increased by 151% compared to the control (rhVEGF-A-untreated HLMEC). The stimulated proliferation of HLMEC by VEGF-A was inhibited 14, 54, and 91% by 1, 2.5, and 5 μM 6,8-DG treatment, respectively (Figure 1B). In our previous paper, we studied the effect of 3-*O*-acetyloleanolic acid on VEGF-A-induced lymphangiogenesis and lymph node metastasis [7]. In the study, the stimulated proliferation of HLMEC by VEGF-A was inhibited 84% by 5 μM 3-*O*-acetyloleanolic acid (a probable positive control of 6,8-DG). This indicates that the effect of 6,8-DG is somewhat greater (7%) than that of 3-*O*-acetyloleanolic acid.

The effect of 6,8-DG on tube formation of rhVEGF-A-treated HMLEC was investigated using a 48-well plate precoated with Matrigel. The tube formation of rhVEGF-A-treated HMLECs was increased by 113% compared to the control. In addition, the increased tube formation of HLMECs by rhVEGF-A was decreased dose-dependently in the presence of 1, 2.5, and 5 μM 6,8-DG (Figure 2A,C). The migration of rhVEGF-A-treated HLMECs was increased by 160% compared to the control. Also, the increased HMLEC migration was reduced by 62, 88, and 104% in the presence of 1, 2.5, and 5 μM 6,8-DG, respectively (Figure 2B,D). These results suggest that 6,8-DG significantly inhibits the VEGF-A-induced proliferation, tube formation, and migration of HLMEC.

### 2.2. Effects of 6,8-Diprenylgenistein on rhVEGF-A-Induced Lymphangiogenesis In Vivo in a Matrigel Plug

We confirmed the anti-lymphangiogenic effect of 6,8-DG in vivo using the in vivo Matrigel plug assay. We performed immunohistochemical analysis using the antibody against LYVE-1, a lymphatic vessel marker, to confirm the lymphatic vessel density in excised plugs. LYVE-1 intensity in the rhVEGF-A-treated Matrigel plugs were increased by 191% compared with the control (rhVEGF-A-treated Matrigel plugs). LYVE-1 intensity of the rhVEGF-A and 6,8-DG-treated Matrigel plug was decreased to 97% compared with the rhVEGF-A-treated Matrigel plug (Figure 3). These results showed that rhVEGF-A stimulates lymphatic vessel formation in vivo, and 6,8-DG inhibits rhVEGF-A-induced lymphatic vessel formation.

### 2.3. Effects of 6,8-Diprenylgenistein on Expression of Lymphangiogenic Factors in CoCl_2_-Treated SCCVII Cells

RT-PCR and Western blot analysis were performed to investigate the effect of 6,8-DG on the expression of lymphangiogenic factors in CoCl_2_-treated SCCVII cells. VEGF-A transcript level was increased 191% in CoCl_2_-treated cells compared to control (CoCl_2_-untreated cells). The increased expression of VEGF-A transcript due to CoCl_2_ was significantly reduced by 31, 54 and 122% in 1, 2.5 and 5 μM 6,8-DG treated cells, respectively. However, the transcript level of VEGF-C, known as a major lymphangiogenic factor, did not change significantly compared to VEGF-A in cells treated with CoCl_2_ in the presence and absence of 6,8-DG. (Figure 4A,B). The VEGF-A protein level was increased 61.9% in CoCl_2_-treated cells compared to control cells. The increased expression of VEGF-A protein was reduced by 70, 89, and 95% in 1, 2.5 and 5 μM 6,8-DG treated cells, respectively. The VEGF-C protein expression did not change significantly compared to VEGF-A in cells treated with CoCl_2_ in the presence and absence of 6,8-DG (Figure 4C,D). These results indicate that the expression of VEGF-A in SCCVII cells increased in response to hypoxia, and 6,8-DG decreases the expression of VEGF-A.

### 2.4. Effects of 6,8-Diprenylgenistein on Activation of VEGF Receptor-2, and Lymphangiogenesis-Related Downstream Signaling Factors in rhVEGF-A-Treated HLMECs

VEGF-A binds to VEGFR-1 and VEGFR-2 on the surface of endothelial cells and leads to phosphorylation of the cytoplasmic domains of VEGFR-1 and VEGFR-2. It leads to activation of lymphangiogenesis-related downstream signaling factors, such as FAK, PI3K, AKT, JNK, ERK, and p38. In order to investigate the effect of 6,8-DG on phosphorylation of VEGFR-1 and VEGFR-2, the tyrosine-phosphorylated proteins were immunoprecipitated with anti-p-Tyr in rhVEGF-A-treated HLMEC lysates and the phosphorylation levels of VEGFR-1 and -2 were determined by Western blot analysis using anti-VEGFR-1 and -2 in immunoprecipitated proteins. The phosphorylation levels of VEGFR-1 and -2 were increased 114% and 138.7% in rhVEGF-A-treated HLMECs compared to control (rhVEGF-A-untreated HLMECs), respectively. The increased phosphorylation of VEGFR-2 was reduced by 28, 103, and 129% in 1, 2.5, and 5 μM 6,8-DG treated HLMECs, respectively. However, the increased phosphorylation of VEGFR-1 was not significantly changed by 6,8-DG (Figure 5A,C). We examined the effects of 6,8-DG on the activation of lymphangiogenesis-related downstream signaling factors: FAK, PI3K, AKT, JNK, ERK, and p38 in rhVEGF-A-treated HLMEC using Western blot analysis. 6.8-DG inhibited the phosphorylation of FAK, PI3K, AKT, JNK, ERK, and p38 induced by rhVEGF-A in rhVEGF-A-treated HLMECs (Figure 5B,D). These results suggest that 6,8-DG inhibits lymphangiogenesis via suppression of the activation of VEGFR-2, and their lymphangiogenesis-related downstream signaling factors.

### 2.5. Effects of 6,8-Diprenylgenistein on Oral Cancer Related Lymphangiogenesis

It has been reported that HIF-1α is an important hypoxia-inducible factor and the expression of lymphangiogenic factors, especially VEGF-A, was stimulated by HIF-1α [28]. The level of HIF-1α expression is mainly determined by transcriptional regulation and proteasome degradation. We confirmed the effect of 6,8-DG on HIF-1α expression using RT-PCR and Western blot analysis in CoCl_2_-treated SCCVII cells. 6,8-DG had little effect on mRNA level of HIF-1α in CoCl_2_-treated SCCVII cells (Figure 6A,B). However, the protein level of HIF-1α was inhibited by 1, 2.5, and 5 μM 6,8-DG in CoCl_2_-treated SCCVII cells (Figure 6C,D). These results show that 6,8-DG does not affect transcriptional regulation of HIF-1α, but is involved in proteasome degradation.

To confirm the effects of 6,8-DG on the interaction between the VEGF-A expressed on cancer cells and VEGFR-1 and -2 of the surfaces of lymphatic endothelial cells, we performed co-immunoprecipitation analysis. VEGF-A bound to VEGFR-1 and VEGFR-2 in the HLMEC lysate to which 6,8-DG was not added. However, in the HLMEC lysate to which 6,8-DG was added, the 6,8-DG did not affect the binding of VEGF-A and VEGFR-1 but it was interfered by the binding of VEGF-A and VEGFR-2 (Figure 6E). Thus, 6,8-DG stimulates the proteasome degradation of HIF-1α in cancer cells, thereby inhibiting the expression of VEGF-A, a major lymphangiogenic factor in oral cancer. Also, 6,8-DG inhibits the binding of VEGF-A and VEGFR-2 in lymphatic endothelial cells (Figure 6F).

### 2.6. Effect of 6,8-Diprenylgenistein on Lymphangiogenesis and Lymph Node Metastasis in a VEGF-A-Induced OCSLN Animal Model

To investigate the in vivo effects of 6,8-DG on VEGF-A-induced lymphangiogenesis and lymph node metastasis, we used a VEGF-A-induced OCSLN animal model, which established VEGF-A overexpressing SCCVII cells (SCCVII/mVEGF-A) in our previous studies [7]. To confirm the presence of tumor cells in the SLNs, we performed H&E staining and an immunohistochemistry analysis using cytokeratin antibody in each group. Cytokeratin is a marker for identifying squamous cell carcinoma cells. We confirmed that SLN metastasis increased due to VEGF-A and the increased SLN metastasis was decreased by 6,8-DG treatment. (Figure 7A). We also confirmed that the volume of the SLN in the SCCVII/mVEGF-A injected group was increased compared to the control (PBS injected) group, from 1.78 ± 0.31 mm^3^ to 21.38 ± 7.6 mm^3^. 6,8-DG inhibited the volume increase of SLN by VEGF-A to 7.61 ± 5.13 mm^3^ (Figure 7B).

To investigate the effects of 6,8-DG on VEGF-A-induced lymphangiogenesis in Tongues (primary tumors) and SLNs, we performed immunohistochemical analyses using anti-LYVE-1 antibody. In the SCCVII/mVEGF-A injected group, LYVE-1 intensity was increased in the tongue and SLN compared to those of the SCCVII injected group. The LYVE-1 intensities of the tongue and SLN in the 6,8-DG treated group were decreased by 96% and 66%, respectively, as compared to the SCCVII/mVEGF-A injected group (Figure 7C,D). These results indicated that 6,8-DG inhibits VEGF-A-induced lymphangiogenesis and SLN metastasis in the OCSLN animal model.

## 3. Discussion

Treatment of oral squamous cell carcinomas (OSCC) involves surgery, radiation, and chemotherapy, and it is usually treated with chemotherapy and radiation after surgery. However, in head and neck cancer, surgery is avoided to preserve the function of the face or organ, and chemotherapy and radiation therapy are mainly used. However, the existing chemotherapy affects not only cancer cells, but also normal cells, resulting in many side effects. Therefore, we attempted to find novel compounds that can inhibit lymph node metastasis of oral cancer among natural compounds derived from edible plants with fewer side effects.

It has been reported that hypoxia conditions are induced in T2 stage oral tongue cancer [29]. CoCl_2_, a chemical hypoxic mimicking agent, induces a hypoxia-like condition similar to that of in vivo tumors by preventing the proteosomal degradation of HIF-1α [30,31]. In this study, we evaluated the in vitro effects of 6,8-DG in SCCVII of hypoxia condition induced by CoCl_2_. To investigate the in vitro inhibitory effects of 6,8-DG, concentrations of 1, 2.5, and 5 μM, which were non-cytotoxic levels in SCCVII and HLMEC, were used. We confirmed the effects of 6,8-DG on the expression of lymphangiogenic factors such as the VEGF family in CoCl_2_-treated SCCVII cells. VEGF-A was most significantly changed by 6,8-DG in CoCl_2_-treated SCCVII cells. VEGF-B, -C, -D, and other lymphangiogenic factors did not change significantly by CoCl_2_ and 6,8-DG in SCCVII cells (data not shown). Thus, VEGF-A is one of the major lymphangiogenic factors in SCCVII cells responding to 6,8-DG under the hypoxic conditions.

Generally, VEGF-A is well known as an angiogenic activator that stimulates angiogenesis by binding to its cell surface receptors, VEGFR-1 and -2, in the endothelial cells of the vascular system [7,32]. In recent studies, it has been reported that VEGF-A stimulates the proliferation and migration of lymphatic endothelial cells, and that overexpression of VEGF-A induces lymphangiogenesis in tumors and sentinel lymph node (SLN)s as well as lymph node metastasis [15]. VEGF-A binds to its receptors and activates signaling factors related to the proliferation and migration of endothelial cells such as PI3K, AKT, and ERK1/2 [7]. In this study, we confirmed that 6,8-DG inhibited the activation of the VEGFR-2 and VEGFR-2 downstream signaling factors involved in the proliferation and migration of HLMECs.

Overexpression of HIF-1α occurs early in the development of oral cancer. In tumor, HIF-1α is known to increase the expression of growth factors such as VEGF-A, -C, and -D and contribute to angiogenesis, lymphangiogenesis, cell survival, invasion and metastasis [33]. In particular, it is known that HIF-1α binds to the hypoxic-response site of the VEGF promoter and regulates VEGF expression [34]. In this study, 6,8-DG inhibited the expression of HIF-1α by stimulating the proteasome degradation of HIF-1α in CoCl_2_-treated SCCVII cells. Although further research is needed, these results show that 6,8-DG probably inhibited the expression of VEGF-A via induction of HIF-1α proteasome degradation in CoCl_2_-treated SCCVII cells.

Several studies have reported that the lymph nodes adjacent to the primary tumor are usually enlarged due to reactive lymphadenopathy and tumor metastasis [2,7]. Also, malignant tumors promote lymphangiogenesis in SLNs before metastasis. Tumor-induced lymphangiogenesis plays an important role in promoting the migration of tumor cells from the primary tumor to the SLN. In a VEGF-A-induced OCSLN animal model, we confirmed that the increase of the volume of sentinel lymph nodes by VEGF-A was reduced by 6,8-DG. 6,8-DG also inhibited VEGF-A-induced lymphangiogenesis around the primary tumor and in the sentinel lymph nodes, and the increased sentinel lymph node metastasis by VEGF-A was inhibited in the 6,8-DG treated group. These data suggest that 6,8-DG inhibits VEGF-A-induced lymphangiogenesis and sentinel lymph node metastasis in an OCSLN animal model.

In conclusion, 6,8-DG inhibited VEGF-A expression, one of the major lymphangiogenic factors, in SCCVII cells, and suppressed the VEGF-A/VEGFR-2 signaling pathway in HLMEC. 6,8-DG inhibits VEGF-A-induced lymphangiogenesis and lymph node metastasis in an OCSLN animal model. Therefore, 6,8-DG could be a potential therapeutic agent for the metastasis prevention and treatment of oral cancer.

## 4. Materials and Methods

### 4.1. Cell Lines and Compound

SCCVII (mouse squamous cell carcinoma) cells were obtained from Dr. Han-Sin Jeong (Samsung Medical Center, Seoul, Korea) and maintained in RPMI-1640 medium (HyClone; GE Healthcare Life Sciences, Logan, UT, USA) added with 10% fetal bovine serum (FBS; HyClone) in a 5% CO_2_ humidified incubator at 37 °C. HLMECs (Lonza, Basel, Switzerland) were maintained in EGM-2 MV bullet kit medium (Lonza) containing 20% FBS in a 5% CO_2_ humidified incubator at 37 °C.

6,8-Diprenylgenistein (6,8-DG, purity more than 98%) was purchased from ChemFaces (cat no. CFN97935, Lot No. CFS201701, Wuhan, Hubei, China). A total of 10 mM stock solution (in DMSO) of 6,8-DG was prepared and protected from light at 4 °C and then diluted as required concentrations in cell culture medium or PBS.

### 4.2. Proliferation, Migration, and Tube Formation Assay

HLMEC proliferation assay performed as follows: HLMEC (5 × 10^4^ cells/well) in EBM-2 (Lonza) supplemented 1% FBS were seeded to each well of gelatinized 24-well plates. After overnight, the culture medium was replaced with EBM-2 containing 20 ng/mL rhVEGF-A (R&D systems, Minneapolis, MN, USA) and/or 6,8-DG (0, 1, 2.5, 5 μM). Cells were incubated for 48 h and were trypsinized and counted using a hemocytometer.

HLMEC migration assay was performed using transwell 24-well plates and inserts with 0.8 μm pore-sized polycarbonate membranes (SPL Life Science, Pocheon, Gyeonggi-do, Korea). HLMEC (5 × 10^4^ cells/well) in EBM-2 containing 6,8-DG (0, 1, 2.5, 5 μM) was added to the upper chamber of the insert. EBM-2 containing rhVEGF-A (20 ng/mL) was added to the lower chamber to induce cell migration. After 24 h, the migrated cells to the underside of inserts were fixed with methanol and stained with hematoxylin solution. Five images per well were taken, and the numbers of migrated cells were counted.

HLMEC tube formation assay performed as follows: 1:1(*v*/*v*) mixture of EBM-2 and growth factor reduced Matrigel (Corning, Glendale, AZ, USA) was added to each well of 48-well plate and allowed to polymerize overnight at 37 °C. HLMEC (5 × 10^4^ cells/well) in EBM-2 containing 1% (*v*/*v*) FBS, rhVEGF-A (20 ng/mL) and/or 6,8-DG (0, 1, 2.5, 5 μM) were added to each well. After 8 h, cells were photographed under an inverted phase contrast microscope using a digital single-lens reflex camera and total tube lengths of a unit area were quantified using the Image J program (NIH, Bethesda, MD, USA, version 1.51j8).

### 4.3. In Vivo Matrigel Plug Assay

A total of 300 μL of Matrigel aliquots containing rhVEGF-A (500 ng/mL) and 6,8-DG (0, 5 μM) were injected bilaterally into the flank area of 5-week-old female BALB/c mice (Orient Bio Inc., Seongnam, Gyeonggi-do, Korea). After 14 days, Matrigel plugs were excised and fixed overnight in 10% neutral buffered formalin, embedded into paraffin and sectioned to a thickness of 5 μm. Paraffin sections were used for immunohistochemical analysis.

### 4.4. Reverse Transcription-Polymerase Chain Reaction (RT-PCR) Analysis

Total RNA was isolated from SCCVII cells using Trizol reagent (Invitrogen, Carlsbad, CA, USA) according to the protocol supplied by the manufacturer. Two μg of total RNA were used for cDNA synthesis with an Improm-^II^ Reverse Transcription System kit (Promega, Madison, WI, USA). The reverse transcription procedure was performed following the manufacturer-provided protocol in a 20 μL reaction mixture containing oligo(dT) primer. PCR products were obtained from Dream taq (Thermo Fisher Scientific Inc., Waltham, MA, USA), and 2 μL of cDNA was used for PCR with specific primers using mouse VEGF-A, 5′-GCCCTGAGTCAAGAGGACAG-3′ (forward) and 5′-GAAGGGAAGATGAGGAAGGG-3′ (reverse); mouse VEGF-C, 5′-CCACAGTGTCAGGCAGCTAA-3′ (forward) and 5′-ACTGCATGTTTGATGGTGGA-3′ (reverse); mouse Hif-1α, 5′-TGACGGCGACATGGTTTACA-3′ (forward) and 5′- AATATGGCCCGTGCAGTGAA-3′ (reverse); and mouse β-actin, 5′-ATGAAACTACATTCAATTCCATCAT-3′ (forward) and 5′-AAACAAAACAATGTACAAAGTCCTC -3′ (reverse). PCR products were resolved on 1% agarose/Tris-acetate EDTA gels that were electrophoresed then visualized with ethidium bromide. PCR product band intensity values were determined using the Image J program (NIH, version 1.51j8).

### 4.5. Western Blot and Immunoprecipitation Analysis

Cells were harvested and lysed with RIPA buffer (Thermo Scientific Inc., Boston, MA, USA) containing a protease inhibitor cocktail (Sigma-Aldrich, St. Louis, MO, USA) and a phosphatase inhibitor cocktail (Sigma-Aldrich). Total protein concentrations were quantified using an RC/DC protein assay reagent (Bio-Rad, Hercules, CA, USA). Protein extracts were separated using 6 and 10% SDS-PAGE and transferred onto PVDF membranes (PALL, Westborough, MA, USA). Membranes were incubated in a blocking solution (Santa Cruz Biotech. Inc., Santa Cruz, CA, USA) for 30 min and incubated with anti-VEGF-A, anti-VEGFR-1, anti-VEGFR-2, anti-phospho FAK, anti-phospho ERK1/2, anti-phospho PI3K, anti-phospho AKT, anti-phospho JNK, anti-phospho p38 antibodies (Santa Cruz Biotech. Inc.) at 1:2000 dilution in a blocking solution overnight at 4 °C, and probed with peroxidase conjugated secondary antibodies at 1:5000 dilution. Protein bands were detected using enhanced chemiluminescent Western blotting detection reagent (Thermo Scientific Inc.)

Also, protein extracts were immunoprecipitated using a mouse anti-phospho-Tyr antibody (Santa Cruz Biotech. Inc.) and an ImmunoCruz™ IP/WB Optima kit (Santa Cruz Biotech. Inc.). Immunoprecipitated proteins were subjected to 6% SDS-PAGE and Western blotting using anti-VEGFR-1 and anti-VEGFR-2 antibodies (Santa Cruz Biotech. Inc.)

### 4.6. ELISA Assay

SCCVII cells were treated with serum-free medium containing 1, 2.5, 5 and 10 μM 6,8-DG and 100 μM CoCl_2_. The conditioned media were collected, 100 μL of conditioned medium was added to a 96-well plate coated with monoclonal antibody against VEGF165 (R&D Systems Inc.) and then incubated for 2 h at room temperature. After three PBS washes, a peroxidase conjugated polyclonal VEGF-A antibody (Santa Cruz Biotech. Inc.) was added, followed by further incubation for 2 h at room temperature. The chromogenic substrate (TMB solution, SurModics, Eden Prairie, MN, USA) was added and the stop solution was added after 30 min. The absorbance of 450 nm was measured using a microplate Reader (Bio-TEK Instruments Inc., Winooski, VT, USA).

### 4.7. Co-Immunoprecipitation Assay

HLMECs were lysed using a RIPA buffer (Thermo Fisher Scientific Inc.) with a protease inhibitor cocktail and a phosphatase inhibitor cocktail. The cell lysates were incubated with the anti-VEGF-A antibody (2 μg), rhVEGF-A (1 μg), and/or 5 μM 6,8-DG at 4 °C overnight. A total of 50 μL of the agarose A/G beads (Santa Cruz Biotech. Inc.) were added and rotated at 4 °C for 3 hr. After washing the beads with cold PBS buffer, proteins were removed from the beads in 30 μL 2× sample buffer and analyzed by 6% SDS-PAGE and Western blotting using anti-VEGFR-1 and anti-VEGFR-2 antibodies (Santa Cruz Biotech. Inc.).

### 4.8. Animals and Study Design

The animal care facility and study protocols (KHUASP-18-006) were approved by the Kyung Hee University Institutional Animal Care and Use Committee. Animal care and experimental procedures were according to the Kyung Hee University guidelines for the care and use of laboratory animals. BALB/c (5 weeks old, female) mice were purchased from ORIENT BIO Inc. Mice received water and food ad libitum while quarantined in a controlled environment with a 12 h light/dark photoperiod.

To establish an OCSLN animal model, 5 × 10^5^ cells of mouse VEGF-A overexpressing SCCVII (SCCVII/mVEGF-A) were submucosally injected into the right side of the tongue of BALB/c mice. Mice were randomly divided into 7 mice per group. Mice of each group were treated with PBS or 6,8-DG (2.5 mg/kg in PBS) every 2 days for 14 days by intraperitoneal injection. All mice were monitored daily for 14 days. To trace lymphatic drainage, 10 μL of 1% Evan’s blue dye was injected into sites of tumor cell inoculation. After 1 h, All Mice were euthanized using CO_2_ gas inhalation, the blue-stained sentinel lymph node was distinguished from other lymph nodes. Tongues (primary tumors) and sentinel lymph nodes were collected in each group of mice. The length and width of the primary tumor and sentinel lymph node were measured using a caliper and the volume was calculated using a standard formula (length × width^2^ × 0.5). The collected tumors and sentinel lymph nodes were fixed overnight in 10% neutral buffered formalin, embedded into paraffin and sectioned to a thickness of 5 μm. Paraffin sections were stained with H&E to visualize histopathologic changes. Paraffin sections were also used for immunohistochemical analysis.

### 4.9. Immunohistochemical Analysis

Paraffin sections of matrigel plugs, primary tumors and SLNs were deparaffinized in xylene, rehydrated in sequentially diluted ethanol, and washed with distilled water. After that, sections were boiled in a 10 mM sodium citrate (pH 6.0) for 10 min. To inhibit the activity of endogenous peroxidase, sections were incubated with methanol containing 1% hydrogen peroxide for 10 min, then blocked with 10% normal serum (Vector Laboratories, Burlingame, CA, USA) for 1 h, followed by incubation overnight in anti-cytokeratin 14 (ab7800, Abcam, Cambridge, UK) and anti-LYVE-1 (ab14917, Abcam) primary antibodies diluted with the blocking solution. Sections were probed with horseradish peroxidase conjugated anti-rabbit IgG antibody, and incubated with DAB solution (Vector Laboratories) until the desired stain intensity developed. After counterstaining with hematoxylin, the sections were examined under the Olympus BX21 inverted microscope (Olympus, Tokyo, Japan). To analyze immunohistochemical signals within specimens, all sections were digitized under 200× objective magnification and images were captured, then analyzed using the Image J program (NIH, version 1.51j8).

### 4.10. Statistical Analysis

Excel in Office 2016 was used to perform the statistical analysis. All data are presented as a mean ± S.D. Student’s *t*-test and one-way ANOVA were used to evaluate the significance between groups (^#^
*p* < 0.01, ^##^
*p* < 0.001: rhVEGF-A only treated group versus the PBS-treated control group, * *p* < 0.05, ** *p* < 0.01, *** *p* < 0.001: rhVEGF-A only treated group versus rhVEGF-A- and 6,8-DG-treated groups).

## Figures and Tables

**Figure 1 ijms-22-00770-f001:**
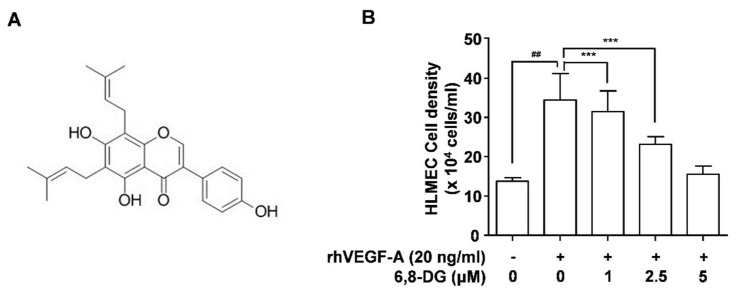
Structure of 6,8-diprenylgenistein (6,8-DG) and effects of 6,8-DG on VEGF-A-induced human lymphatic microvascular endothelial cells (HLMEC) proliferation. (**A**) Structure of 6,8-DG. (**B**) Proliferation in HLMECs stimulated rhVEGF-A. Cells were detached and counted using a hemocytometer. Data are presented as a mean ± S.D. of three independent experiments (^##^
*p* < 0.001, *** *p* < 0.001).

**Figure 2 ijms-22-00770-f002:**
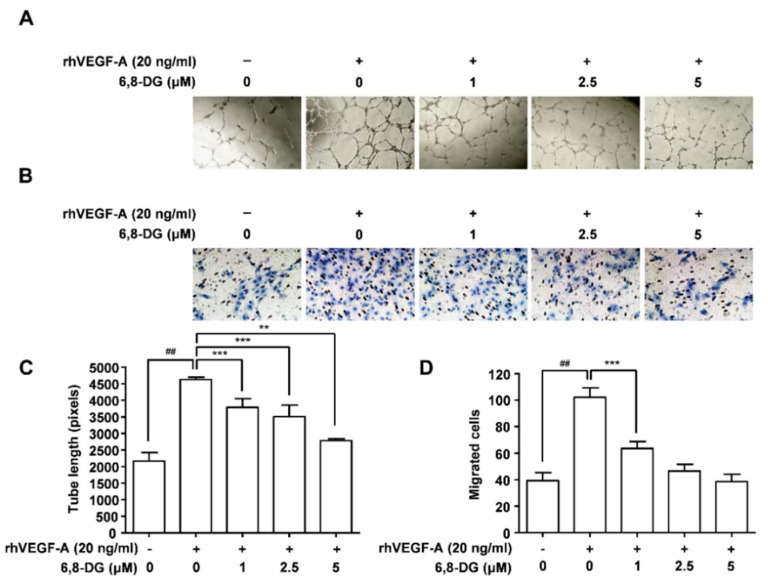
Effects of 6,8-DG on VEGF-A-induced HLMEC tube formation and migration. (**A**,**C**) Tube formation in HLMECs stimulated rhVEGF-A. Cells were imaged under an inverted phase contrast microscope using a digital single-lens reflex camera. Total tube lengths of a unit area were calculated using the Image J program. (**B**,**D**) The migrated cells to the underside of membranes were fixed with methanol, stained with hematoxylin solution, and then imaged under an inverted phase contrast microscope using a digital camera. Five digital images per well for (**B**) were obtained, and the numbers of migrated HLMECs were counted. Each sample was assayed in duplicate. Numbers of migrated HLMECs present in 320 mm^2^ are presented as a bar diagram. Data are presented as a mean ± S.D. of three independent experiments (^##^
*p* < 0.001, ** *p* < 0.01, *** *p* < 0.001).

**Figure 3 ijms-22-00770-f003:**
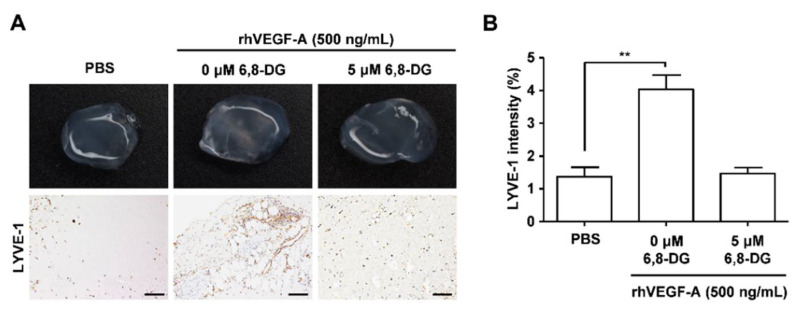
Effects of 6,8-DG on VEGF-A-induced lymphangiogenesis in an in vivo Matrigel plug. (**A**) Matrigel plugs were excised and photographed using a digital camera. The lymphatic vessel density values in Matrigel plug sections were measured using the immunohistochemical analysis with anti-LYVE-1 antibody. All Matrigel sections were digitalized and microscopic images were captured under 200× objective magnification. Scale bar = 200 μm. (**B**) Immunohistochemical intensity values (LYVE-1) from captured images were analyzed by the Image J program and represented as a bar diagram. Data are presented as a mean ± S.D. (** *p* < 0.01).

**Figure 4 ijms-22-00770-f004:**
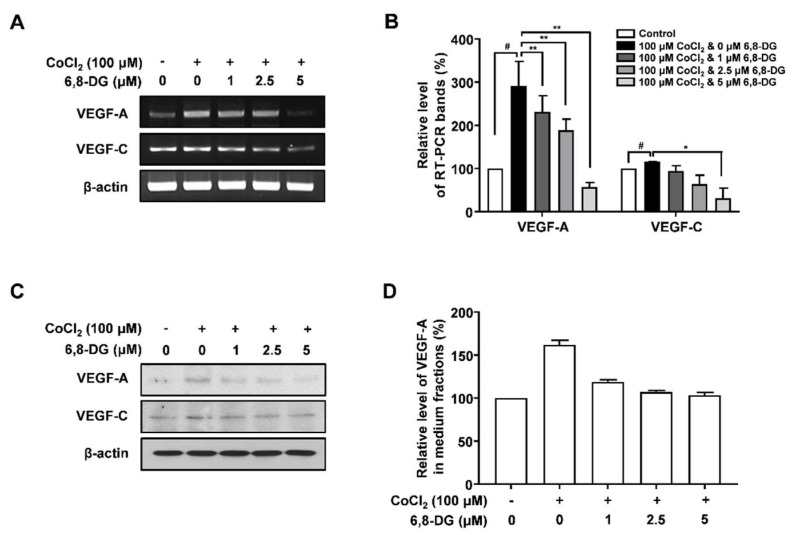
Effects of 6,8-DG on the expression of VEGF-A and VEGF-C in SCCVII cells treated with CoCl_2_. (**A**) cDNAs were generated from total RNAs treated with DNase I, and PCR reaction was performed with specific primers of VEGF-A and -C and β-actin. (**B**) PCR products from three independent experiments (**A**) were quantified and represented as a bar diagram. The levels of the VEGF-A and -C transcripts in the control (6,8-DG- and CoCl_2_-untreated cells) were estimated as 100%. (**C**) The protein levels of VEGF-A and -C in the intracellular fraction were determined using Western blot with anti-VEGF-A and anti-VEGF-C antibodies. (**D**) The protein level of VEGF-A in medium fractions was determined using ELISA assay. Data are presented as a mean ± S.D. of three independent experiments (^#^
*p* < 0.01, * *p* < 0.05, ** *p* < 0.01).

**Figure 5 ijms-22-00770-f005:**
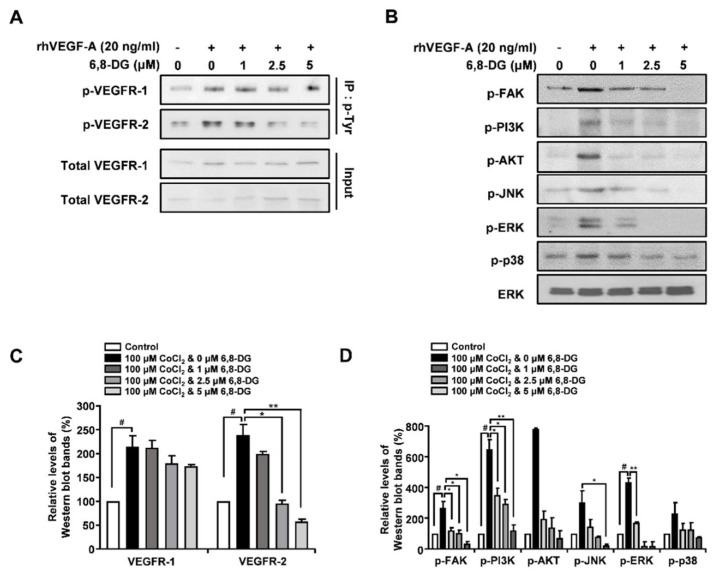
Effects of 6,8-DG on activation of VEGFR-1, VEGFR-2 and lymphangiogenesis-related downstream signaling factors in rhVEGF-A-treated HLMECs. (**A**,**C**) Cell lysates were immunoprecipitated with anti-phospho-Tyr (anti-p-Tyr). The levels of phosphorylated VEGFR-1 and -2 in immunoprecipitates were detected using Western blot analysis with anti-VEGFR-1 and anti-VEGFR-2 antibodies. (**B**,**D**) HLMECs were serum starved for 6 h, then were treated with different concentrations of 6,8-DG (0,1, 2.5, 5 μM) in the presence of rhVEGF-A (20 ng/mL) for 30 min. The phosphorylation levels of FAK, PI3K, AKT, JNK, ERK1/2, and p38 were determined using Western blot analysis with anti-p-FAK, anti-p-PI3K, anti-p-AKT, anti-p-JNK, anti-p-ERK1/2, and anti-p-p38 antibodies. Data are presented as a mean ± S.D. of three independent experiments (^#^
*p* < 0.01, * *p* < 0.05, ** *p* < 0.01).

**Figure 6 ijms-22-00770-f006:**
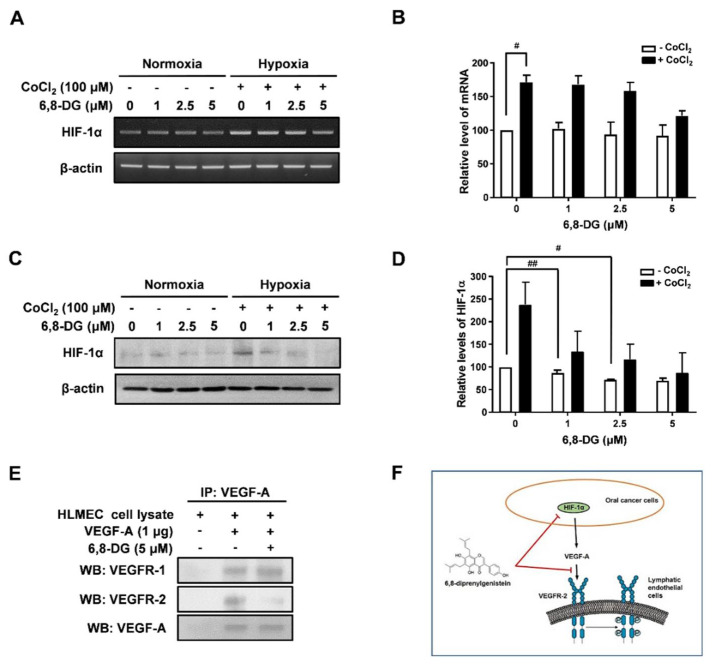
Effects of 6,8-DG Effect of 6,8-DG in oral cancer related lymphangiogenesis. (**A**) SCCVII cells were treated with different concentrations of 6,8-DG (0, 1, 2.5, 5 μM) in the presence or absence of 100 μM of CoCl_2_ for 24 h. cDNAs were generated from total RNAs treated with DNase I, and PCR reaction was performed with specific primers of HIF-1α and β-actin. (**B**) PCR products from three independent experiments (**A**) were quantified and represented as a bar diagram. The levels of the VEGF-A and -C transcripts in the control (6,8-DG- and CoCl_2_-untreated cells) were estimated as 100%. (**C**,**D**) The cells were lysed and samples subjected to SDS-PAGE. Western blot was performed with anti-HIF-1α and anti-β-actin antibodies. (**E**) Total protein extracts of HLMECs were collected and analyzed by co-immunoprecipitation with rhVEGF-A and anti-VEGF-A antibody. The protein levels of VEGFR-1, VEGFR-2, and VEGF-A were detected by Western blot using anti-VEGFR-1, anti-VEGFR-2, and anti-VEGF-A antibodies. (**F**) Schematic illustration of the mechanism of action of 6,8-DG in oral cancer related lymphangiogenesis. Data are presented as a mean ± S.D. of three independent experiments (^#^
*p* < 0.01, ^##^
*p* < 0.001).

**Figure 7 ijms-22-00770-f007:**
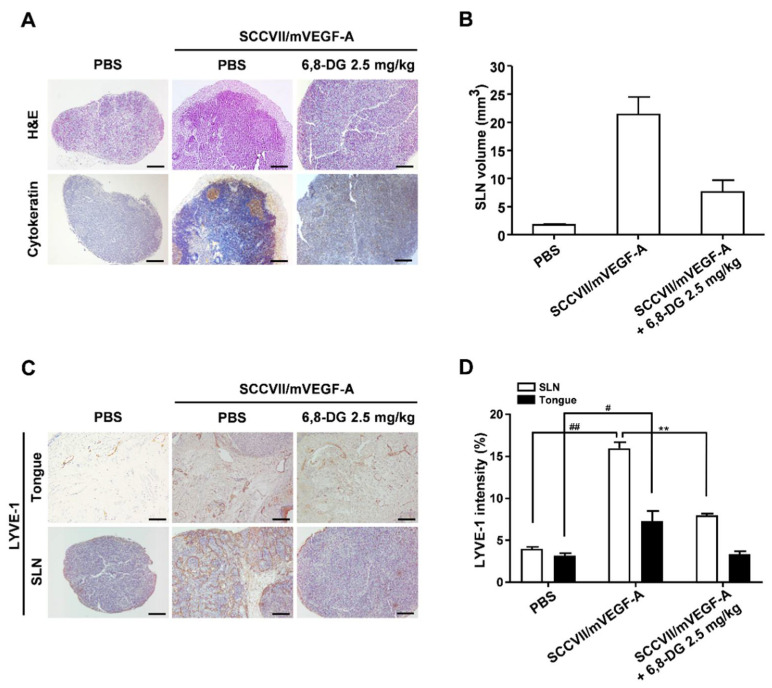
Effects of 6,8-DG on lymphangiogenesis and lymph node metastasis in a VEGF-A-induced OCSLN animal model. (**A**) Sentinel lymph nodes sections of mice of each group were analyzed using hematoxylin and eosin staining, and immunohistochemical analysis with anti-cytokeratin antibody. (**B**) Sentinel lymph nodes volumes were measured using a caliper. (**C**) Lymphatic vessel density values in sentinel lymph nodes sections were measured by immunohistochemical analysis using anti-LYVE-1 antibody. All sections were digitalized and images were captured under 200× objective magnification. Scale bar = 200 μm. (**D**) Immunohistochemical intensity values of LYVE-1 from captured images of tumors and sentinel lymph nodes were analyzed via the Image J program and represented as a bar diagram. Data are presented as a mean ± S.D. (^#^
*p* < 0.01, ^##^
*p* < 0.001, ** *p* < 0.01).

## Data Availability

All data generated or analyzed during this study are included in this published article.

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
