# Peer review of "Effects of 6,8-Diprenylgenistein on VEGF-A-Induced Lymphangiogenesis and Lymph Node Metastasis in an Oral Cancer Sentinel Lymph Node Animal Model"

_ijms, 2021, doi:10.3390/ijms22020770_

Round 1

Reviewer 1 Report

In this study the authors have investigated the Effects of 6,8-Diprenylgenistein on VEGF-A-induced Lymphangiogenesis and Lymph Node Metastasis in an Oral Cancer Sentinel Lymph Node Animal Model.

I would suggest:

To modify the abstract structuring the different parts (Introduction, Materials and methods etc).

To provide new images in particular of histology and immunohistochemisty (ie  figure 2A and 3 (LYVE)-1 (they are not of good quality).

Immunohistochemical analysis : it should be precised what antibody was used for keratin labelling. To give information on the control used (isotype used etc).

Reference should be uniformized for style.

Author Response

In this revision, we have addressed all the comments from the reviewers. The ‘Author to respond reviewer’ has been included as an attachment file.

The cover letter for our revision is as follows.

Dear Editor:

Thanks to you and the reviewers for the valuable comments. We are submitting the revised manuscript (ijms-1075218) entitled “Effects of 6,8-Diprenylgenistein on VEGF-A-induced Lymphangiogenesis and Lymph Node Metastasis in an Oral Cancer Sentinel Lymph Node Animal Model”. In this revision, we have addressed all the comments from the reviewers. Also, we have highlighted all the changes to our manuscript using colored text.

I hope our revision meets your approval and look forward to hearing from you.

Sincerely yours,

In Sik Chung

Reviewer 2 Report

In the manuscript entitled “Effects of 6,8-Diprenylgenistein on VEGF-A-induced Lymphangiogenesis and Lymph Node Metastasis in an Oral Cancer Sentinel Lymph Node Animal Model”, the authors illustrated that 6,8-DG could be novel and valuable therapeutic agents for metastasis prevention and treatment of oral cancer. The manuscript is clearly and tightly written and the authors' conclusions are well supported by the data presented. There are a few comments that should be addressed to further improve the manuscript.

Major:

  1. The authors should include both loss-of-function and gain-of-function experiments to validate the effects of 6,8-DG on the interaction between the VEGF-A and HIF-1α.
  2. What are the 6,8-DG functions in the normal cells and other cancer cell lines?

Minor:

  1. There are no GAPDH or b-actin in Fig 5B. The analysis of Fig 5D needs to be more clear.
  2. Fig 6F can be moved to Fig 7E.

Author Response

In this revision, we have addressed all the comments from the reviewers. The ‘Author to respond reviewer’ has been included as an attachment file.

The cover letter for our revision is as follows.

Dear Editor:

Thanks to you and the reviewers for the valuable comments. We are submitting the revised manuscript (ijms-1075218) entitled “Effects of 6,8-Diprenylgenistein on VEGF-A-induced Lym-phangiogenesis and Lymph Node Metastasis in an Oral Cancer Sentinel Lymph Node Animal Model”. In this revision, we have addressed all the comments from the reviewers. Also, we have highlighted all the changes to our manuscript using colored text.

I hope our revision meets your approval and look forward to hearing from you.

Sincerely yours,

In Sik Chung

Round 2

Reviewer 2 Report

This work is interesting and the results are also beneficial for identifying 6,8-DG as a therapeutic agent for metastasis prevention and treatment of oral cancer. No further comment on the revised manuscript.